Molecular characterisation of the Chlamydia pecorum plasmid from porcine, ovine, bovine, and koala strains indicates plasmid-strain co-evolution

Jelocnik Martina 1
Bachmann Nathan L. 1
Seth-Smith Helena 2
Thomson Nicholas R. 3
Timms Peter 1
Polkinghorne Adam M. 1 apolking@usc.edu.au
1 Centre for Animal Health Innovation, University of the Sunshine Coast , Sippy Downs, Queensland , Australia
2 Functional Genomics Center Zurich, University of Zurich , Zurich , Switzerland
3 Infection Genomics, The Wellcome Trust Sanger Institute , Cambridge , United Kingdom
Decaro Nicola
Electronic publication date: 2016 Feb 4
Publication date: 2016
Volume: 4
Electronic Location ID: e1661
Received 2015 Nov 2; Accepted 2016 Jan 14
Copyright: ©2016 Jelocnik et al.
Copyright year: 2016
Copyright holder: Jelocnik et al.
License: This is an open access article distributed under the terms of the Creative Commons Attribution License, which permits unrestricted use, distribution, reproduction and adaptation in any medium and for any purpose provided that it is properly attributed. For attribution, the original author(s), title, publication source (PeerJ) and either DOI or URL of the article must be cited.
License URL: https://creativecommons.org/licenses/by/4.0/

Keywords: Chlamydia pecorum, Molecular characterisation, Comparative analyses, Plasmid, Phylogeny, Koala

Funding: ARC Discovery Project DP130102066 Queensland Government Department of Environment and Heritage Protection Koala Research Grant KRG18 Wellcome Trust grant #098051 This work was funded by an ARC Discovery Project grant (DP130102066) awarded to PT and AP and a Queensland Government Department of Environment and Heritage Protection Koala Research Grant (KRG18) awarded to AP and PT. NRT is supported by a Wellcome Trust grant #098051 from the Wellcome Trust Sanger Institute. The funders had no role in study design, data collection and analysis, decision to publish, or preparation of the manuscript.

==============================
Background. Highly stable, evolutionarily conserved, small, non-integrative plasmids are commonly found in members of the Chlamydiaceae and, in some species, these plasmids have been strongly linked to virulence. To date, evidence for such a plasmid in Chlamydia pecorum has been ambiguous. In a recent comparative genomic study of porcine, ovine, bovine, and koala C. pecorum isolates, we identified plasmids (pCpec) in a pig and three koala strains, respectively. Screening of further porcine, ovine, bovine, and koala C. pecorum isolates for pCpec showed that pCpec is common, but not ubiquitous in C. pecorum from all of the infected hosts.

Methods. We used a combination of (i) bioinformatic mining of previously sequenced C. pecorum genome data sets and (ii) pCpec PCR-amplicon sequencing to characterise a further 17 novel pCpecs in C. pecorum isolates obtained from livestock, including pigs, sheep, and cattle, as well as those from koala.

Results and Discussion. This analysis revealed that pCpec is conserved with all eight coding domain sequences (CDSs) present in isolates from each of the hosts studied. Sequence alignments revealed that the 21 pCpecs show 99% nucleotide sequence identity, with 83 single nucleotide polymorphisms (SNPs) shown to differentiate all of the plasmids analysed in this study. SNPs were found to be mostly synonymous and were distributed evenly across all eight pCpec CDSs as well as in the intergenic regions. Although conserved, analyses of the 21 pCpec sequences resolved plasmids into 12 distinct genotypes, with five shared between pCpecs from different isolates, and the remaining seven genotypes being unique to a single pCpec. Phylogenetic analysis revealed congruency and co-evolution of pCpecs with their cognate chromosome, further supporting polyphyletic origin of the koala C. pecorum. This study provides further understanding of the complex epidemiology of this pathogen in livestock and koala hosts and paves the way for studies to evaluate the function of this putative C. pecorum virulence factor.

Introduction

Chlamydial plasmids are often referred to as virulence plasmids. They are small, highly conserved, non-integrative and non-conjugative plasmids that are not known to carry genetic cargo such as antibiotic resistance genes. Still, these enigmatic plasmids appear to be an essential part of the genome of the majority of species in the family Chlamydiaceae, with plasmids almost ubiquitously found in human-specific species such as C. trachomatis as well as in those that infect animals such as C. psittaci, C. caviae, C. felis, C. muridarum, and C. pneumoniae (Harris et al., 2012; Nunes & Gomes, 2014; Pickett et al., 2005; Read et al., 2013; Rockey, 2011; Thomas et al., 1997). The exceptions to the above are: (a) the related integrated plasmid in C. suis, which has been shown to be able to carry a tetracycline resistance (tetC) gene (Dugan et al., 2004); (b) C. abortus, which is not known to possess any plasmids (Sait et al., 2011) and; (c) very rare human naturally occurring plasmid-free C. trachomatis isolates (Peterson et al., 1990; Stothard et al., 1998).

An increasing number of studies have linked the chlamydial plasmid to the pathogenic potential of an infecting isolate, as well as to disease outcome. In animal models, it was demonstrated that naturally occurring plasmid-free C. trachomatis and/or plasmid-cured C. muridarum isolates were less infective and less virulent than the wild type plasmid positive ones (Russell et al., 2011; Sigar et al., 2014). Further, C. muridarum studies also demonstrated the critical role of the plasmid in the development and severity of intrauterine infections (Chen et al., 2015; Liu et al., 2014). The utility of the chlamydial plasmid encoded proteins has also been explored as targets for vaccine and diagnostic test development. Plasmid-cured C. trachomatis strains have shown potential as live attenuated vaccines against ocular chlamydial infections in primate models, by providing a complete protection against challenge by a virulent plasmid bearing strain (Kari et al., 2011). For diagnostic purposes, the plasmid secreted Pgp3 protein has been explored as a marker of chlamydial infections for both humans and animals (Donati et al., 2009; Li et al., 2008), while the C. trachomatis specific plasmid sequence was used as a target for commercial molecular diagnostic test for the C. trachomatis infections. However, the emergence of new Swedish C. trachomatis variants with deletion in the plasmid target sequence evaded diagnostics (Ripa & Nilsson, 2007), highlighting importance of the knowledge of chlamydial plasmid sequences and structure (Seth-Smith et al., 2009).

To date, evidence for a C. pecorum plasmid has been scarce. Recent ovine, bovine, and koala C. pecorum whole genome sequencing studies did not report the presence of a plasmid in any of the resulting genome sequences (Bachmann et al., 2014; Bachmann et al., 2015; Mojica et al., 2011; Sait et al., 2014). In our latest comparative genomics study of C. pecorum from a variety of hosts, we identified four complete C. pecorum plasmids (pCpec) in the genomes of a porcine and three koala C. pecorum isolates, using a set of nine available previously sequenced porcine, ovine, bovine and koala C. pecorum genome data sets (Jelocnik et al., 2015a). Sequence analysis showed that all four pCpecs were 7.5 kbp in length with eight predicted CDSs with 99% nucleotide sequence identity, and an overall nucleotide sequence identity of 67–70% to orthologous genes from chlamydial plasmids in different species (Jelocnik et al., 2015a). A subsequent PCR-based pCpec screening of 114 C. pecorum strains from pigs, sheep and cattle, and 113 strains from koalas revealed that pCpec, while present in C. pecorum taken from all of the infected hosts, is not ubiquitous in all C. pecorum isolates: pCpec was detected in 38.6% of the sampled C. pecorum taken from livestock, while pCpec was more commonly detected in the Australian koala C. pecorum isolates, with a 55.8% detection rate (Jelocnik et al., 2015a). This varying pCpec distribution in koala and livestock C. pecorum strains potentially sets this plasmid apart from those described in other chlamydial species.

In the current study, we fully sequenced and characterised 17 novel pCpecs from a set of 16 C. pecorum strains from the most common hosts of this pathogen (pigs, sheep, cattle and koalas) in order to examine the genetic structure and diversity of pCpec. In doing so, we observed that, although conserved, the pCpec sequences are genetically diverse, while the pCpecs phylogenies indicated congruency and co-evolution with its cognate C. pecorum chromosome.

Methods and Materials

C. pecorum positive clinical samples and isolates used for pCpec characterisation and analyses

A total of 17 novel pCpec were characterised from a small collection of cultured C. pecorum isolates and C. pecorum positive clinical swab/tissue samples from two porcine, four ovine, three bovine, and seven koala hosts, previously screened positively for pCpec presence. C. pecorum samples used in the present study and their descriptions are outlined in Table 1.

Table 1 Descriptions of C. pecorum samples used for plasmid characterisation.

Plasmid (pCpec) ID	Host and country of origin	Type of sample/ Anatomical site	Host pathology	Plasmid generation	Length (bp)	GC content (%)	Accession number	Strain reference	
L1*	Pig, Austria	Culture/Lung	Pneumonia	WGSc	7,548	31.7	KT223773	Koelbl (1969)	
R106	Pig, Austria	Culture/Lung	Pneumonia	Amplicon seq.d	7,548	31.7	KT223776	Koelbl (1969)	
1886	Pig, Austria	Culture/Lung	Pneumonia	Amplicon seq.d	7,548	31.7	KT223767	Koelbl (1969)	
IPA	Sheep, USA	Culture/Joint	Polyarthritis	Amplicon seq.d	7,547	31.6	KT223771	Bachmann et al. (2014)	
W73	Sheep, Ireland	Culture/Faeces	Asymptomatic	Amplicon seq.d	7,547	31.6	KT223780	Sait et al. (2014)	
Cur/E19/Rec	Sheep, Australia	Swab sample/Rectum	NCDa	Amplicon seq.d	7,547	31.6	KT223769	Jelocnik et al. (2014b)	
Cur/E11/Rec	Sheep, Australia	Swab sample/Rectum	NCDa	Amplicon seq.d	7,547	31.6	KT223768	Jelocnik et al. (2014b)	
LW623	Cattle, USA	Culture/Brain	Encephalomyelitis	Amplicon seq.d	7,547	31.6	KT223774	Kaltenboeck, Kousoulas & Storz (1993)	
WA/B31/Ileal	Cattle, Australia	Tissue sample/Ileum	SBEb	Amplicon seq.d	7,548	31.7	KT223781	Jelocnik et al. (2014a)	
66P130	Cattle, USA	Culture/Faeces	NCDa	Amplicon seq.d	7,548	31.7	KT223766	Kaltenboeck, Kousoulas & Storz (1993)	
SA/K84/Ure	Koala, Australia	Swab sample/Urethra	NCDa	Amplicon seq.d	7,547	31.6	KT223778	Jelocnik et al. (2015a)	
SA/K09/Ure	Koala, Australia	Swab sample/Urethra	NCDa	Amplicon seq.d	7,547	31.6	KT223777	Jelocnik et al. (2015a)	
Vic/R6/UGT	Koala, Australia	Swab sample/UGT	NCDa	Amplicon seq.d	7,547	31.6	KT223779	Jelocnik et al. (2015a)	
Marsbar*	Koala, Australia	Culture/UGT	Cystitis	WGSc	7,547	31.5	KT223775	Bachmann et al. (2014)	
IPTaLE*	Koala, Australia	Culture/Ocular	Conjunctivitis	WGSc	7,547	31.5	KT223772	Bachmann et al. (2014)	
DBDeUG*	Koala, Australia	Culture/UGT	UGT infection	WGSc	7,547	31.5	KT223770	Bachmann et al. (2014)	
HazBoEye	Koala, Australia	Culture/Ocular	Conjunctivitis	WGSc	7,547	31.8	KT352920	Jelocnik et al. (2015a)	
HazBoUGT	Koala, Australia	Culture/UGT	Conjunctivitis	WGSc	7,547	31.8	KT352921	Jelocnik et al. (2015a)	
NoHerEye	Koala, Australia	Culture/Ocular	Conjunctivitis	WGSc	7,547	31.5	KT352922	Jelocnik et al. (2015a)	
TedHUre	Koala, Australia	Culture/Urethra	Cystitis	WGSc	7,547	31.5	KT352923	Jelocnik et al. (2015a)	
PMHaUre	Koala, Australia	Culture/Urethra	Cystitis	WGSc	7,547	31.5	KT352924	Jelocnik et al. (2015a)	
Notes.

* Previously characterised plasmid.

a No clinical disease.

b Sporadic bovine encephalomyelitis.

c Contig from whole genome sequencing.

d Conventional PCR overlapping fragments, dideoxy sequenced.

Chlamydial sequences used for phylogenetic analyses in the present study

In the present study, we have also used publicly available: (i) plasmid sequences of C. pneumoniae pLPColN (NC017286); C. muridarum pMoPn (AE015926); C. trachomatis pCTA (CP000052); C. avium p10DC88 (CPOO6571); C. felis pCfe1 (AP006862); C. psittaci p6BC (CP002550); C. caviae pCpGP 1 (AE015926); and C. pecorum pCpecL1 (KT223773), and (ii) 16S rRNA gene sequences of the corresponding plasmid-bearing chlamydial strains: C. pneumoniae LPCoLN (FJ236984); C. muridarum MoPn (CP007217); C. trachomatis A/HAR-13 (NR025888); C. avium 10DC88 (NR121781); C. felis Fe/C56 (NC007899); C. psittaci 6BC (NR102492); C. caviae GPIC (NR036833); and C. pecorum L1 (LFRH01000000) for sequence and the Bayesian phylogenetic analyses.

pCpec characterisation from C. pecorum whole genome sequencing data

The raw Illumina MiSeq reads of five unpublished koala C. pecorum genomes (HazBoEye, HazBoUGT, NoHerEye, TedHUre, and PMHaUre), sequenced at Wellcome Trust Sanger Institute, Cambridge, UK, were mapped to the newly identified pCpecL1, and pCpecMarsbar sequences (Jelocnik et al., 2015a). Reads mapping to pCpecL1 or pCpecMarsbar were assembled from the data sets of five koala C. pecorum isolates. The resulting assemblies described a ∾7.5 kbp plasmid, each composed of a single contig (see Table 1). Assembled plasmids were aligned with the porcine pCpecL1 and koala pCpecMarsbar using ClustalX, as implemented in Geneious 7.1.4 (Kearse et al., 2012).

Primers design for pCpec amplicon sequencing

Using the pCpecL1 and koala pCpec sequences, we designed 23 oligonucleotide PCR primers to amplify overlapping pCpec fragments (Fig. 1 and Table S1). Primers were tested for DNA base mismatches using Basic Local Alignment Search Tool (BLAST) (http://blast.ncbi.nlm.nih.gov/Blast.cgi#), as well as analysed in OligoAnalyzer 3.1 online tool (https://sg.idtdna.com/calc/analyzer). Primers were designed to have similar annealing temperatures so that they could be used in various combinations to amplify products of between ∼700 bp and 3.4 kbp fragments (e.g., PG6 For and PG8 Rev: 3.4 kbp fragments; PG3 For and PGP3 Rev: 731 bp; Table S1).

Figure 1 Graphical representation of the pCpecL1 and annotated CDSs, including primer locations.

Putative ori at the top. The 22 bp tandem repeat units are indicated by blue arrows.

A subset of 12 C. pecorum positive DNA samples (Table 1), obtained from various anatomical sites from porcine, ovine, bovine, and koala hosts were used for further plasmid identification and characterisation. These samples were part of the 227 sample collection previously screened for pCpec presence (Jelocnik et al., 2015a), by our pCpec-specific PCR. PCR amplifications for pCpec fragments up to 1 kbp were performed as previously described (Jelocnik et al., 2015a), with appropriate annealing temperature (Table S1). For the amplification of fragments (>1.5 kbp) the LongRange PCR kit, Qiagen, Victoria, Australia, was used as per manufacturer instructions. Isolated pCpecMarsbar, pCpecIpTaLe, pCpecDbDeUG DNA from koala C. pecorum strains MC/Mars, DBDeUG, and IpTaLE were used as positive controls. Genomic DNA from the non-plasmid containing strains L17 and L71, and dH20 were used as negative controls (Jelocnik et al., 2015a). The presence of the amplicons were confirmed on 1.5% agarose-TBE gels, and then purified and dideoxy sequenced (The Institute for Future Environments (IFE), Queensland University of Technology (QUT), Brisbane, Australia using the Applied Biosystems ABI3500 Gene analyser).

Forward and reverse chromatogram of each sequenced amplicon was aligned in Geneious 7.1.4 and the amplicon consensus sequence was extracted. Overlapping amplicon sequences were used to assemble the full length plasmid sequence for each sample. The derived pCpec sequences were annotated with RAST (Aziz et al., 2008),and further curated in Geneious 7.1.4. pCpecs translated CDSs were further analysed in blastp for comparison (http://blast.ncbi.nlm.nih.gov/Blast.cgi), as well as Universal Protein Resource (UniProt from http://www.uniprot.org/) and Conserved Domains Database (CDD from http://www.ncbi.nlm.nih.gov/cdd) to assess the protein functionality. Plasmid sequences were deposited in Genbank under accession numbers KT223766– KT223781, and KT352920, KT352921, KT352922, KT352923and KT352924.

21 pCpec sequence and phylogenetic analyses

In order to assess the evolutionary relationships of pCpecs and the level of sequence diversity, we determined the number of synonymous (ds) and non-synonymous (dn) substitutions found in all 21 C. pecorum plasmid sequences included in this study. The number of polymorphic (segregating) sites, CDS alleles, plasmid genotypes, and putative recombination events were determined using DnaSP 5.0 (Librado & Rozas, 2009). A pCpec ancestral sequence was reconstructed using an alignment of all 21 pCpecs performed on the FastML server (Ashkenazy et al., 2012). Best-fit models of nucleotide substitution used for phylogenetic analysis of the plasmid sequences were estimated by jModelTest v.2.1.1 (Darriba et al., 2012). A rooted Bayesian phylogenetic tree consisting of eight chlamydial plasmids or their corresponding 16SrRNA gene sequences were constructed with MrBayes as implemented in Geneious 7.1.4, with HKY + I + G for plasmid, and HKY + G for 16S rRNA sequences. Both trees used the C. muridarum plasmid or 16S rRNA sequences as a root. A Bayesian phylogenetic tree consisting of all 21 C. pecorum 7.5 kbp plasmid sequences was also constructed with MrBayes using GTR + G model. C. pneumoniae plasmid pLPCoLN sequence was used as an outgroup. The plasmid phylogenetic tree was compared to the chromosome Multi Locus Sequence Typing (MLST-based) phylogeny constructed from the concatenated sequence of the seven MLST house-keeping (HK) gene fragments (Jelocnik et al., 2013) for corresponding samples, where available (excluding SA/K84/Ure, SA/K09/Ure and Vic/R6/UGT ). A Bayesian MLST phylogenetic tree of 18 C. pecorum samples sequences was also constructed in MrBayes with the HKY85 + I + G model. Run parameters for all phylogenetic trees from this study included four Markov Chain Monte Carlo (MCMC) chains with a 1,000,000 generations, sampled every 100 generations and with the first 10,000 trees were discarded as burn-in.

Results and Discussion

pCpec: a newly characterised member of chlamydial plasmids

With the recent observation that the C. pecorum plasmid is common but not ubiquitous across this species (Jelocnik et al., 2015a), we characterised an additional 17 novel pCpec sequences from C. pecorum strains isolated from a variety of hosts. In total, we included a set of these 21 C. pecorum plasmid sequences for analyses.

These data revealed that all 21 pCpecs were 7.5 kbp in size with a low G + C (av. 31.6%) content, typical for chlamydial plasmids (Thomas et al., 1997). Using an alignment of eight representative plasmid sequences from related chlamydial species (including pCpec), we constructed a plasmid phylogeny and compared it to the 16S rRNA gene phylogeny of the corresponding strains carrying these plasmids (Fig. S1). As observed in Fig. S1 and consistent with previous phylogenetic analysis of chlamydial plasmids from other species (Mitchell et al., 2010), plasmid phylogenies displayed a similar topology to that observed for the 16S rRNA derived chlamydial phylogeny. In this study, both C. pecorum pCpecL1 and the C. pecorum 16SrRNA-gene sequence clustered with its closest chlamydial relative C. pneumoniae (Gupta et al., 2015). The observed phylogenetic relationships of the chlamydial plasmids also suggest that the plasmids evolve in parallel with their bacterial host (Rockey, 2011), and therefore that these plasmids were acquired early in the evolution of Chlamydiae and have been subject to little between-species recombination (Andersson & Kurland, 1998), although intra-species plasmid-associated recombination has been reported previously in C. trachomatis (Harris et al., 2012).

pCpec phylogeny

Overall, the alignment of the 21 pCpec sequences resolved 12 distinct plasmid sequence types (genotypes), with five genotypes shared between pCpecs from different isolates, and the remaining seven genotypes were unique to a single pCpec (Fig. S2). The three porcine pCpecs were of an identical sequence type (Genotype L). Among the 11 koala pCpecs, we identified five distinct genotypes, with: (i) pCpecs SAK09Ure, SAK84Ure and VicR6UGT sharing the first (Genotype A); (ii) pCpecs NoHerEyes, PMHaUre, TedHUre and DbDeUG sharing the second (Genotype B) and; (iii) pCpecs HazBoEye and HazBoUgt sharing the third genotype (Genotype C); in contrast, the pCpecs Marsbar, and IpTale were of a distinct fourth and fifth genotype each (Genotypes D and E, respectively). With the exception of the pCpecs CurE11Rec, and CurE19Rec which were also of an identical sequence type (Genotype F), the remaining two ovine (W73 and IPA), and the three bovine (WAB31Ileal, 66P130, and LW623) pCpecs were of a unique genotype each (Genotypes G, H, I, J and K, respectively) (Fig. S2).

Figure 2 Bayesian phylogenetic analyses of (A) 7.5 kbp 21 pCpec sequences from C. pecorum strains from porcine, ovine, bovine, and koala hosts; and (B) concatenated sequences of the seven MLST C. pecorum genes from 18 corresponding strains harbouring plasmids.

Posterior probabilities >0.75 are displayed on the tree nodes, while the hosts are indicated by the colouring on the legend.

We also compared the phylogenies for our pCpec sequences (Fig. 2A) to the corresponding MLST phylogenetic tree for the strains harbouring these plasmids (Fig. 2B). In the absence of whole genome sequences for a number of isolates, MLST-derived phylogenies were utilised, as they have previously been shown to be congruent with those constructed from core genome alignments (Bachmann et al., 2014; Bachmann et al., 2015). Using the C. pneumoniae plasmid sequence as an out-group, the root of the C. pecorum plasmid tree falls between two distinct pCpec clades (Fig. S3). Further, the phylogenetic tree also resolved 21 pCpec sequences into lineages according to their genotype and/or closely related genotypes (Figs. 2A and S2B). Clade 1 included a distinct lineage consisting of pCpecs from South Australian (SA) and Victorian (Vic) koala strains of an identical genotype, as well as the more diverse lineage that consisted of the bovine pCpec66P130 and pCpecWAB31Ileal, and porcine pCpecs L1, 1886, and R106 sequences (Fig. 2A). The genetically diverse Clade 2 included all the remaining eight pCpecs from Queensland (QLD) and New South Wales (NSW) koala C. pecorum strains forming a well-supported lineage, as well as the pCpecs CurE11Rec, CurE19Rec from Australian sheep strains, and pCpecLW623, pCpecW73 and pCpecIPA from USA bovine and ovine C. pecorum isolates, respectively (Fig. 2A). pCpec phylogeny was found to be largely congruent with a C. pecorum MLST phylogeny (Fig. 2B). In the MLST tree, we also observed two distinct clades. The first clade included lineages from porcine and bovine C. pecorum (Fig. 2B), while the second genetically diverse clade included all the koala C. pecorum strains resolved into their own lineages, and the four ovine (W73, CurE11Rec, CurE19Rec, and IPA) and the bovine LW623 MLST sequences (Fig. 2B).

Overall, this phylogenetic analysis revealed similar levels of inter-host C. pecorum strain genetic variability to that described previously using various other molecular markers (Jelocnik et al., 2015b; Mohamad et al., 2014), as well as congruency with our previously published C. pecorum core genome phylogenies (Bachmann et al., 2014; Jelocnik et al., 2015a). The co-evolution of plasmids with the chromosome of C. pecorum is consistent with that previously described for C. trachomatis (Seth-Smith et al., 2009). Separation of the pCpec sequences into two distinct clades with different lineages suggests that, like the pathogen itself, the current C. pecorum plasmids are of a polyphyletic origin. Using pCpecs from koala strains as an example, we observed an evolutionary split between the genetically identical plasmids from South Australian (SA) and Victorian (Vic) koala C. pecorum strains, and the more genetically diverse plasmids from Queensland (QLD) and New South Wales (NSW) koala strains. It should be also noted that the plasmids from QLD and NSW koala strains appear to be more closely related to the plasmids from sheep strains than to plasmids from strains from the same host, an observation that is consistent with our previous molecular typing and comparative genomics studies of these C. pecorum strains (Bachmann et al., 2015).

Sequence analyses of pCpecs and its encoded proteins

Manually curated annotation of the 21 pCpec sequences revealed a similar structure to that of chlamydial plasmids from other species with eight CDSs predicted in total. CDSs 1, 2, 3, 7 and 8 are putatively involved in plasmid maintenance and replication, while CDSs 4, 5 and 6 are associated with chlamydial-specific virulence (Gong et al., 2013; Song et al., 2013; Thomas et al., 1997) (Table 2). pCpec also carry the four 22 bp tandem repeats located in the putative origin of replication (Figs. 1 and S2A).

Table 2 Characteristics of plasmid CDSs from 21 characterised pCpec from C. pecorum pig, cattle, sheep, and koala strains.

Plasmid CDSs/ annotation	Predicted function	Length (bp)/ predicted a.a	No. of non- synonymous substitutions	dn*	No. of synonymous substitutions	ds*	dn∕ds	Δnt	No. of alleles	
CDS 1/pGP8	Integrase	936/312	2	0.00113	5	0.00995	0.113	7	4	
CDS 2/pGP8	Integrase	1,026/342	6	0.00278	7	0.01280	0.217	13	8	
CDS 3/pGP1	Replicative DNA helicase	1,374/458	4	0.00148	7	0.01130	0.131	11	6	
CDS 4/pGP2	Virulence plasmid proteina	1,026/342	2	0.00074	10	0.01881	0.039	12	6	
CDS 5/pGP3	Virulence plasmid proteina	795/265	4	0.00290	6	0.01219	0.131	10	6	
CDS 6/pGP4	Virulence plasmid proteina	309/103	0	0	3	0.01654	0	3	4	
CDS 7/Par A	Plasmid partitioning protein	783/261	2	0.00097	8	0.01838	0.053	10	4	
CDS 8/pGP6	Plasmid replication protein	744/248	2	0.01839	7	0.00141	0.077	9	6	
Intergenic region (between CDSs 8 and 1)	Origin of replication	207/4 × 22 bp tandem repeats	n.a	n.a	n.a	n.a	n.a	2	3	
Notes.

a Chlamydia specific.

* ds and dn, the average number of synonymous substitutions per synonymous site and non-synonymous substitutions per non-synonymous site, respectively (Jukes—Cantor corrected); Δnt, No. of polymorphic sites; No. of alleles, No. of unique sequences of each CDS.

The alignment of the 21 full length pCpec sequences derived either from genome sequencing or PCR-based approaches, revealed from 0 to 83 SNPs (a maximum split between the two pCpecs sequences) distributed evenly around the plasmid. A single insertion of 1 bp was seen in bovine 66P130 and LW623, and porcine L1, 1886 and R106 pCpecs located in the intergenic region between CDS 6 and CDS 7 (Fig. S2A). Using the reconstructed plasmid sequence as a reference, we also identified pCpec homoplasic SNPs. As observed in the pCpec SNPs-only alignment (Fig. S2B), we detected 10 homoplasic SNPs, three of them (at positions 9, 10 and 84) in the intergenic regions, while the remaining seven (at positions 1, 16, 29, 69, 75, 76 and 78) were in the pCpec CDSs 1, 2, 3, 7 and 8. Homoplasic SNPs at positions 1, 16, 76 and 78 also resulted in a non-synonymous amino acid change (Fig. S2B). Although the most parsimonious explanation is for putative recombination between these strains, in our present analyses using a set of 21 pCpecs we do not have an evidence of recombination, hence it is uncertain whether the observed homoplasic SNPs occurred as a result of a homologous recombination or by independent selection (Harris et al., 2012).

We further assessed each of the eight CDSs as well as the intergenic regions for synonymous and non-synonymous SNPs (Table 2). Most SNPs were detected in the CDS 2 (putative integrase) and CDS 4 (putative plasmid virulence protein), with 13 and 12 SNPs each, respectively. pCpecs CDS 2 also had the most non-synonymous changes (n = 6) (Fig. S2A) and could be detected in eight different alleles. In contrast to the seven polymorphic CDSs with seven to 13 SNPs each, pCpecs CDS 6 was the most conserved sequence, with only three synonymous SNPs.

The level and distribution of the pCpec sequence variation was comparable to that in C. trachomatis plasmids (Ferreira et al., 2013; Seth-Smith et al., 2009). Using only a limited set of pCpec sequences, in our analyses the sequence variation mostly appears to be a result of synonymous changes. Nevertheless, it was interesting to note that most of the non-synonymous changes, including the ones in virulence-associated CDSs 4 and 5, seem to be accumulating in pCpecs from C. pecorum strains derived from sheep, and QLD and NSW koalas presenting with disease (Fig. S2A). In regards to the plasmid virulence-associated CDSs, CDS 4 was the most polymorphic locus with only two non-synonymous changes, however. C. pecorum CDS 5 (pgp3) had the most non-synonymous changes (n = 4), mainly accumulating in the 5’end of the gene, while the CDS 6 was the most conserved. C. trachomatis CDS 5, encoding PgP3, is the most polymorphic CDS in this plasmid (Ferreira et al., 2013; Seth-Smith et al., 2009). Cytosol-exported PgP3 has been implicated as one of the major chlamydial factors affecting disease pathogenesis in C. trachomatis (Li et al., 2008), therefore possibly due to the host’s immune pressure, this CDS is under pressure to accumulate SNPs. At present, it is unknown what selective pressures are being placed on the Chlamydia-specific virulence-associated pCpec CDS 4 and 5 genes. Future studies to investigate this will require a larger set of pCpecs from strains isolated from a variety of diseased as well as healthy hosts, using both bioinformatics as well as cell biology approaches.

Detection of the most variation and non-synonymous changes in pCpec CDS 2 contrasts with its strong conservation in the C. trachomatis plasmid. The remaining pCpec CDSs displayed high sequence conservation, particularly for CDS 7 and CDS 8. These CDSs are putatively denoted as the partitioning co-transcribing genes (Ferreira et al., 2013), and their encoded proteins may play essential role in effective plasmid segregation to the progeny cells, hence their sequence conservation is not surprising. Based on our findings, pCpec sequences appear to be evolutionary conserved and it is likely that the function of the encoded proteins is similar to that previously predicted for C. trachomatis (Seth-Smith et al., 2009), and C. pneumoniae (Thomas et al., 1997).

Conclusions

In the present study, we have characterised the genetic structure and phylogenetic relationships of 21 pCpec from porcine, ovine, bovine, and koala strains. Our data suggests that the pCpec sequence is evolutionarily conserved, as it is in related chlamydial species, with the chlamydial plasmid virulence-associated features present. The pCpecphylogenies revealed “co-evolution” of plasmids with their respective C. pecorum chromosomes, further supporting a polyphyletic evolution of this pathogen at least in Australian koalas (Fig. 2). Based on the pCpec sequence and predicted gene functions, the level and nature of the plasmid conservation suggests that pCpec, where found, is potentially important for during growth, infection, and/or transmission of the bacterium within a population, as has been suggested in studies comparing plasmid-positive and plasmid-negative chlamydial isolates (Rockey, 2011; Russell et al., 2011). Although how this relates to the plasmid-negative C. pecorum is not clear at present. This study provides more clues to understand the complex epidemiology of this pathogen in livestock and koala hosts.

Supplemental Information

Figure S1 Bayesian phylogenetic analyses of plasmid sequences from eight related chlamydial species, compared to the 16S rRNA gene sequences from corresponding chlamydial strains harbouring these plasmids

Bayesian phylogenetic analyses of (A) plasmid sequences from eight related chlamydial species, compared to the (B) 16S rRNA gene sequences from corresponding chlamydial strains harbouring these plasmids. Posterior probabilities >0.75 are displayed on the tree nodes. C. muridarum sequences were used as an out-group. Associated plasmid and 16S rRNA gene sequence from the same chlamydial strain are denoted by coloured arrows.

Click here for additional data file.

Figure S2 pCpec SNP distribution and SNP phylogeny

(A) SNP distribution in the pCpec genotypes, using Genotype A as a reference. SNP positions are highlighted in black, while the type of variants are highlighted in purple for A, pink for G, green for C, light blue for T. SNPs resulting in non-synonymous changes are indicated with red boxes. A single bp insertion in the pCpec genotypes I, J and L are indicated with a green box. The 22bp tandem repeat units are indicated by blue arrows. (B) The pCpec phylogeny aligned to the tracks of pCpec SNP only alignment, using reconstructed plasmid sequence N1 as a reference. Above the alignment is the graphical representation of pCpec CDSs position in reference to the SNPs alignment, while the top line is numbering the successive SNPs as detected in the pCpec sequences. SNPs are highlighted as disagreements to the reference sequence. Homoplasic SNPs are denoted with star symbols. Ones resulting in a non-synonymous change are denoted with red stars, while the ones resulting in a synonymous change are denoted with blue stars.

Click here for additional data file.

Figure S3 Neighbor-Joining phylogenetic analyses of the 21 pCpec sequences from C. pecorum strains from porcine, ovine, bovine, and koala hosts using C. pneumoniae pLPCoLN as an out-group

Neighbor-Joining phylogenetic analyses of the 21 pCpec sequences from C. pecorum strains from porcine, ovine, bovine, and koala hosts using C. pneumoniae pLPCoLN as an out-group. Bootstrap values (1,000 times repetitions) are displayed on the tree nodes.

Click here for additional data file.

Table S1 Primers used for generating plasmid (pCpec) fragments and their characteristics

Click here for additional data file.

We thank Alyce Taylor-Brown for technical assistance and Tamieka Fraser and Courtney Waugh for sample provision.

Additional Information and Declarations

Competing Interests

Author Contributions

Data Availability

The authors declare there are no competing interests.

Martina Jelocnik conceived and designed the experiments, performed the experiments, analyzed the data, contributed reagents/materials/analysis tools, wrote the paper, prepared figures and/or tables, reviewed drafts of the paper.

Nathan L. Bachmann performed the experiments, analyzed the data, prepared figures and/or tables, reviewed drafts of the paper.

Helena Seth-Smith performed the experiments, analyzed the data, contributed reagents/materials/analysis tools, wrote the paper, prepared figures and/or tables, reviewed drafts of the paper.

Nicholas R. Thomson analyzed the data, contributed reagents/materials/analysis tools, wrote the paper, reviewed drafts of the paper.

Peter Timms conceived and designed the experiments, wrote the paper, reviewed drafts of the paper.

Adam M. Polkinghorne conceived and designed the experiments, analyzed the data, wrote the paper, reviewed drafts of the paper.

The following information was supplied regarding data availability:

Plasmid sequences were deposited in Genbank under accession numbers KT223766– KT223781, and KT352920, KT352921, KT352922, KT352923and KT352924.

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
