# Peer review of "Molecular characterisation of the Chlamydia pecorum plasmid from porcine, ovine, bovine, and koala strains indicates plasmid-strain co-evolution"

_PeerJ, doi:10.7717/peerj.1661_

## Round 0.1 · original submission · Minor Revisions

Dear corresponding author,

Both reviewers raised minor points that should be addressed before final acceptance of your manuscript.

Reviewer 1 ·

Basic reporting

Unfortunately, I was not able to find the legends of supplementary figures in the manuscript. That made it difficult to understand and evaluate the manuscript itself. So, please indicate me the legends of supplementary figures.

Line 210-222: I can't evaluate this paragraph without the legend of Figure S2.

Line 266-267: I can't find any tandem repeats in Figure S1 and S2A. Is it indicated in the legend?

Line 270-272: I can't find a single insertion of 1 bp in Figure S2A. Is it indicated in the legend?

Line 276-277: Figure 2B -> Figure S2B?

Line 253-259: I understand SA and Vic koala strains are "SaK09Ure, SaK84Ure" and VicR6UGT, respectively. However, I think there is no specific information about QLD and NSW koala strains in the manuscript.

Experimental design

No comments.

Validity of the findings

Line 238-243: Actually, both pCpecs and C. pecorum strains are divided into two major clades.
However, as far as I referred the study regarding the co-evolutional relationship between plasmids and chromosomes of C. trachomatis (Seth-Smith et al., 2009), I don't think pCpec phylogeny is congruent with C. pecorum MLST phylogeny. Please clarify my reservation.

Additional comments

In the latest study of these authors (Jelocnik et al., 2015b), they identified four plasmids in the genomes of a swine and three koala Chlamydia pecorum isolates. That was the first report regarding plasmid of C. pecorum, resulting pCpec. In this study, they additionally identified 17 novel pCpecs of C. pecorum from a variety of hosts by bioinformatic and PCR-based approaches.
They carried out phylogenetic analysis of the 21 pCpecs and suggested that co-evolution of pCpecs with their host, C. pecorum.

I think that this may be an important step forward to understand the epidemiology of C. pecorum in livestock and koala.

·

Basic reporting

From previously sequenced Chlamydia pecorum genomes and by using a PCR specific of C pecorum plasmid, the authors identified 21 plasmids from bovine, koala, ovine, and porcine strains and characterized their genetic structure and their phylogenetic relationship.
This short note is clear, concise and well written

Experimental design

The scientific content of this article is stimulating. The aim of the work is clearly specified, the used methods are adapted to this objective

Validity of the findings

The conclusions are pertinent. However the discussion on the 3 CDS (CDS4, CD5, CDS6) associated with chlamydial-specific virulence is a little too short. It would be interesting to compare them depending on whether they come from strains isolated from animals with or without clinical signs.

Additional comments

Very nice work!

---

## Round 0.2 · accepted · Accept

The revised manuscript is now suitable for publication.